# Antibiotic Prescribing in Primary Care for Urinary Tract Infections (UTIs) in Pregnancy: An Audit Study

**DOI:** 10.3390/medsci8030040

**Published:** 2020-09-17

**Authors:** Flavia Ghouri, Amelia Hollywood

**Affiliations:** School of Pharmacy, University of Reading, Reading RG6 6DZ, UK; f.ghouri@reading.ac.uk

**Keywords:** antibiotic prescribing, urinary tract infection, pregnancy, audit

## Abstract

Urinary tract infections (UTIs) are associated with negative pregnancy outcomes and are treated with antibiotics. Although beneficial, antibiotic use causes antimicrobial resistance (AMR), and therefore their use needs to be carefully balanced. Antimicrobial guidelines are developed to facilitate appropriate prescribing of antibiotics. This study assessed antibiotic prescribing for UTIs in pregnancy against the National Institute for Health and Care Excellence (NICE) guideline NG109. Fifty antibiotic prescribing records dated from 1st October 2018 to 1st July 2019 were identified from three London-based GP practices. The results show that a mid-stream sample of urine, which is important for the review and tailoring of antibiotic treatment, was collected in 77.6% of cases. Prescribing the first-line antibiotic is important for adequate treatment and good antimicrobial stewardship and results show that 44% of prescriptions were for the first-choice antibiotic. Most prescriptions (56%) were for a second-line or non-recommended antibiotic. Providing self-care advice is key to empowering pregnant women in managing their own health but only 16% of records documented provision of self-care advice. This study highlights important areas of concern in the management of UTIs in pregnancy. However, due to the retrospective design, future work is needed to evaluate the role of AMR in the prescriber’s treatment decision-making process.

## 1. Introduction

Urinary tract infections (UTIs) are caused by bacteria that colonise and infect the urinary tract. The infection commonly causes symptoms such as increased frequency of urination and burning pain when passing urine [1]. More severe symptoms can include nausea, vomiting, chills and a high fever [2]. Bacteria that cause a UTI are usually commensal within the body and normally transfer to the urinary tract from the gut [3]. UTIs are more common in women than in men, due to physiological differences, and estimates in the literature suggest that approximately 50% of women experience at least one episode of a UTI during their lifetime [4]. A shorter urethra in women makes it easier for bacteria from the gut to pass into the urinary tract and cause an infection. UTIs are classified as asymptomatic bacteriuria (presence of bacteria in the urine without symptoms), acute cystitis (affecting the bladder) or pyelonephritis (affecting the kidneys) [5]. 

UTIs are caused by bacteria and therefore antibiotic treatment is usually required to eradicate the infection. Asymptomatic infections in non-pregnant adults should not be medically treated, unless they are undergoing genitourinary surgery, whereas treatment of symptomatic UTIs consists of taking a short course of an antibiotic [6] such as nitrofurantoin or trimethoprim. In pregnancy, however, both symptomatic and asymptomatic UTIs are linked with adverse outcomes such as pyelonephritis. UTIs in pregnancy have also been associated with risks such as foetal growth retardation and pre-term birth [7,8]. They are the most common type of infection affecting pregnant women, with asymptomatic infections alone estimated to affect between 2 and 12% of pregnant women [9]. Clinical vigilance is therefore exercised in pregnancy through routine screening for bacteriuria at early antenatal appointments to detect and treat both symptomatic and asymptomatic UTIs with antibiotics [10,11]. The prescribed duration of the antibiotic course is also longer in pregnancy compared with non-pregnant women to ensure adequate eradication of the infection. 

While antibiotics are highly effective at treating UTIs, they are also associated with antimicrobial resistance (AMR), which is a global health threat. AMR is the term used to describe the evolution of microorganisms to develop characteristics that make them resistant to antimicrobial treatment [12]. Antibiotic prescribing and consumption are directly associated with a rise in bacterial resistance and the development of antibiotic resistant infections [13]. Resistant infections can be difficult to treat and are associated with significant morbidity and mortality [14]. There is evidence to suggest that antibiotics to treat UTIs are overused in pregnant women [15,16], which can cause a rise in resistant UTIs. Resistant UTIs can be particularly concerning in pregnancy [17] because although they can be a challenge to treat in general, ensuring the combined safety of the woman and foetus adds further difficulty in pregnancy. Considering this, it is necessary to monitor the use of antibiotics in pregnancy to optimise prescribing and consumption of these valuable medicines and facilitate antimicrobial stewardship in antenatal care. 

Primary care antibiotic prescribing during pregnancy has been studied in the UK and shows that UTIs accounted for the highest proportion of antibiotics in pregnant women [18]. The National Institute for Health and Care Excellence (NICE) published antimicrobial prescribing guidelines which recommend evidence-based treatment for infections to promote the judicious use of antibiotics. The present study was conducted to identify and assess the appropriateness of antibiotic prescribing for UTIs in pregnancy in line with the NICE antimicrobial prescribing guidelines for lower UTI [19]. The aim of the study was to assess the prescribing of antibiotics for UTIs in pregnant women. 

## 2. Materials and Methods 

The GP practices that were included in this study are part of a group in London that provide primary care services in the community. Three practices from this group were sufficient to achieve the required sample size. Collectively, the three practices have a total of 16 GPs, a team of pharmacy professionals and approximately 35,000 registered patients [20] from a range of ethnic backgrounds. The surgeries provide services that are typically representative of other practices in the area. The Chief Pharmacist, who is the Clinical Director at the health group, approved the study and permitted the researchers to access the premises and retrieve patient data. An ethics application was also submitted prior to conducting the study and was reviewed by the University of Reading School of Chemistry, Food and Pharmacy Ethics Committee and was given favourable ethical opinion for conduct (Study 12/19). The study used consecutive sampling to identify records of women who had been prescribed antibiotics while they were pregnant. The inclusion criteria were women who were pregnant and had been prescribed an antibiotic indicated for a UTI. A sample size of 50 sequential patient records was set as the target sample size; the rationale for this is that it is in line with a similar audit conducted previously [15]. Medical notes were assessed to collect data on antibiotic prescribing, and determine the appropriateness of the prescriptions according to the audit standards which were adapted from the NICE antimicrobial guidelines for lower UTI [19]. The standards are as follows: A midstream urine sample (MSSU) is obtained and sent for culture before antibiotics are taken.The choice of antibiotic prescribed is reviewed based on culture results and changed as appropriate to a narrow spectrum antibiotic.
First choice antibiotic: nitrofurantoin 100 mg modified release twice a day (or 50 mg four times a day) for 7 days.If no improvement within 48 h or if first line is unsuitable:Second choice: Amoxicillin 500 mg three times a day for 7 days or cefalexin 500 mg twice a day for 7 days. Provision of self-care advice on managing pain and maintaining adequate hydration.

A data collection tool was designed to collect demographic, clinical and treatment data from the medical notes. Table 1 summarises the key information that was collected using the data collection tool. 

FG was given an induction of the SystmOne^©^ GP prescribing software and trained on accessing patient medical notes prior to data collection. A search was conducted to retrieve medical notes of pregnant women using the read code for ‘pregnancy’, which was then combined with a search for antibiotics that are normally used for UTIs in the UK, regardless of pregnancy status. These antibiotics include nitrofurantoin, amoxicillin, cefalexin, which are recommended for UTIs in pregnancy, and trimethoprim, co-amoxiclav, fosfomycin and pivmecillinam, which are not recommended but can be used to treat UTIs in non-pregnant cases. Data were collected in July 2019 and the search criteria were limited between the dates of 1st October 2018 to 1st July 2019 to correspond with the publication of the NICE guidelines. The medical notes were examined to confirm that the antibiotics were for the treatment of a UTI during pregnancy. The medical notes were also used to establish the appropriateness of deviating from the guidelines’ recommendation, e.g., the medical notes were consulted when a second line antibiotic was prescribed to determine whether it was because of contraindications to the first-line choice. Information from the medical notes was then documented on the data collection tool. Each of the 50 individual records were assigned a number to prevent identification and protect the anonymity of patients. The results were collated on Microsoft Excel^®^ and are presented using descriptive statistics which summarises the proportion of prescriptions that met each standard as a percentage. 

## 3. Results

The target sample size for the study was to assess fifty patients’ records, which was achieved through consecutive sampling of data from the three GP practices. The records were reviewed, and information was recorded using the data collection form. The mean age of the women was 31.1 years (±4.3) and there was a range of ethnicities attending the practices, as seen in Table 2.

### 3.1. Standard 1: A Midstream Urine Sample (MSSU) Was Obtained and Sent for Culture before Antibiotics Are Taken

A mid-stream urine sample (MSSU) was collected before antibiotics were prescribed in 38/49 (77.6%) of cases. An MSSU was not collected in 11/49 (22.4%) of instances and documentation was unclear in 1/50 (0.02%) prescriptions. 

### 3.2. Standard 2: The Choice of Antibiotic Prescribed Was Reviewed Based on Culture Results and Changed as Appropriate to a Narrow Spectrum Antibiotic

First choice antibiotic: nitrofurantoin 100 mg modified release twice a day (or 50 mg four times a day) for 7 days.If no improvement within 48 h or if first line was unsuitable:Second choice: Amoxicillin 500 mg three times a day for 7 days or cefalexin 500 mg twice a day for 7 days.

Nitrofurantoin is the first choice of antibiotic according to the NICE guidelines unless there is a known contraindication to it—for example, the woman is in the third trimester of pregnancy or is allergic to nitrofurantoin. Nitrofurantoin was prescribed in 19/50 (38.0%) cases. Amoxicillin and cefalexin are the second-choice antibiotics which should be used if nitrofurantoin is contraindicated. These two antibiotics were prescribed in 27/50 (54.0%) of cases. Assessment of the medical notes and culture results revealed that 3/27 (11.1%) were prescribed amoxicillin or cefalexin because nitrofurantoin was contraindicated, therefore these were deemed as appropriate and meeting the guidelines. Trimethoprim and co-amoxiclav were two antibiotics which are not recommended for UTIs in pregnancy but were prescribed in 4/50 (8.0%) of cases. Therefore, the overall proportion of prescriptions that met the guidelines in terms of the first choice for empirical antibiotic therapy was 22/50 (44.0%). The proportion of prescriptions which are included in the guidelines for UTI treatment but are not the recommended as first choice was 24/50 (48.0%). The proportion of prescriptions which did not meet the guidelines completely was 4/50 (8.0%). The antibiotics prescribed according to the trimester of pregnancy can be seen in Table 3.

The dose and frequency of the antibiotic were according to the guidelines in 38/46 (82.6%) of prescriptions whereas 8/46 (17.4%) were prescribed a dose that did not meet the guidelines, as shown in Table 4.

The NICE guidelines recommend a seven-day course of antibiotics in pregnancy and 32/50 (64.0%) complied with this recommendation whereas 18/50 (36.0%) had a shorter duration (3 or 5 days) prescribed. Overall, 25/50 (50%) of antibiotic prescriptions did not meet the NICE guidelines either in terms of the choice of antibiotic and/or the dose or treatment duration that was prescribed. 

### 3.3. Standard 3: Provision of Self-Care Advice on Managing Pain and Maintaining Adequate Hydration

This standard was considered to have been met if the medical notes mentioned provision of any advice on analgesics and/or emphasized maintaining adequate hydration. Self-care advice was provided in 8/50 (16.0%) occasions, but there was no documentation for the provision of self-care advice in the medical notes for 42/50 (84.0%) records.

### 3.4. Additional Clinical Data 

Clinical data on the allergy status, whether the UTI was symptomatic or asymptomatic and the trimester of pregnancy were also collected. There were 5/50 (10.0%) instances where women were allergic to antibiotics, which included beta lactams and trimethoprim. Symptomatic infections accounted for 38/50 (76.0%) of UTIs treated, whereas asymptomatic infections occurred in 7/50 (14.0%) cases. The type of infection was not documented in 4/50 (8.0%) of cases. Due to screening for asymptomatic bacteriuria in the first trimester [21], it was expected that the overall incidence of UTIs would be highest in this period. However, the largest incidence of UTIs seen in this study was in the second trimester 21/50 (42.0%) followed by the first 18/50 (36.0%) and the third 11/50 (22.0%) trimesters.

## 4. Discussion

The study assessed 50 antibiotic prescriptions for UTIs in women who were pregnant. The women were from a range of ethnic backgrounds and at various stages of pregnancy in terms of trimesters. The following discussion highlights important criteria where the prescribing of antibiotics for UTIs in pregnancy did not meet the NICE antimicrobial guidelines for lower UTIs and the potential implications of deviating from recommended practice. These criteria correspond to the audit standards and include the collection of a MSSU, the choice of empirical antibiotic, the duration of the course and the provision of self-care advice.

The NICE guideline recommends the collection of a MSSU sample to test the susceptibility of the causative bacteria prior to the use of antibiotics. Culturing a MSSU sample is important as it can not only confirm the diagnosis of a UTI but also allows for the review and tailoring of the treatment by ensuring that the prescribed antibiotic is effective based on the antibiotic sensitivity of the causative organism [22]. Empirically prescribed antibiotics should be reviewed after the culture results are returned, which is typically within 48 h. Although an MSSU was obtained in the majority of cases, 22.4% of women were prescribed antibiotics without obtaining a urine sample, without a justifiable reason. Without a culture result, prescribers cannot review the choice of antibiotic prescribed which can lead to unnecessary and prolonged use of multiple antibiotics along with undermining antimicrobial stewardships efforts in primary care. 

Nitrofurantoin is the first-choice antibiotic recommended by NICE as well as the European Association of Urology [23]. It is the first choice for treatment of UTIs based on evidence of its effectiveness [24] and because it has one of the lowest rates for resistance of E.coli [19], which is the most common uropathogen causing UTIs [11]. It is, however, contraindicated in pregnancy if the woman is at term, in the third trimester, because of the risk of neonatal haemolysis. It should also be avoided if renal function is impaired or if there is a known allergy or intolerance to it [25,26]. The second-choice antibiotic prescriptions recommended by NICE are amoxicillin and cefalexin. Both of these antibiotics are broad-spectrum with amoxicillin belonging to the aminopenicillin class, which has high rates of *E. coli* resistance globally [27]. Due to antimicrobial resistance, these antibiotics are not preferred as a first-line option but instead are used if symptoms do not improve after using the first-choice antibiotic for a minimum of 48 h or if the first choice cannot be used because of contraindications. The results show that amoxicillin and cefalexin were often prescribed as the empirical first choice despite the patient being suitable for nitrofurantoin. While clinically this may lead to the intended outcome of clearing the infection, it is not best practice when considering the broader problem of AMR. The medical notes indicate that a possible explanation for the preference of amoxicillin or cefalexin might be due to their perceived safety in pregnancy compared with nitrofurantoin, but this needs to be confirmed and addressed through future work. 

The most concerning antibiotic prescriptions identified in this study were trimethoprim and co-amoxiclav. These are problematic because trimethoprim is a folate antagonist and unsafe in pregnancy, and short courses of co-amoxiclav have low effectiveness for treating UTIs [28]. The high proportion of second-line and non-recommended antibiotic prescribing is an important finding because they do not represent the optimal choice in terms of antimicrobial stewardship or antibiotic safety in pregnancy. This result highlights that the choice of antibiotic is an important area where a change in prescribing requires a review. It is therefore recommended that a review of prescribers’ perceptions of antimicrobial safety in pregnancy is conducted to elucidate their views to justify and inform future interventions.

The duration of antibiotic treatment in pregnancy is recommended to be seven days compared with three days in non-pregnant women. The results show that the course length was not long enough for 36% of prescribed antibiotics. There is no concrete evidence to suggest that a seven day course of antibiotics is essential [29,30] and the advice for patients to complete an antibiotic course has also been questioned [31]. However, traditionally it has been thought that shortening courses may not provide complete eradication of the infection. Therefore, unless new evidence emerges through randomised clinical trials, it is important that the course length of antibiotics is in line with the recommended seven days for adequate treatment. 

Provision of self-care advice is a criterion where there was very low compliance with NICE guidelines. Comprehensive self-care advice includes several hygiene behaviours such as drinking adequate water, avoiding delay to use the toilet and wiping the genital area from front to back. However, in this study, self-care measures incorporated advice on suitable painkillers and emphasis on maintaining adequate hydration. Previous research shows that women require and value advice given by their GP for pain management in cystitis [32]. The result from this study shows that self-care advice was only reported in 16% of consultations. The importance of self-care advice lies not only in the empowerment of patients to manage their own condition but also in the protective role that preventative behaviours, such as drinking water, can provide. Preventative behaviours are associated with a reduced incidence of UTIs in pregnancy [33], and are therefore key in minimising reliance on antibiotics. Therefore, it is recommended that healthcare professionals emphasise these in consultations with women. An evidence-based intervention leaflet that encourages shared decision-making and self-care for uncomplicated UTIs has been developed [34]. It has also been endorsed by NICE [35] and contains helpful material that could be used in consultations with pregnant women to encourage preventative behaviours. 

### 4.1. Future Work 

Studies exploring the perceptions of women who experienced UTIs in pregnancy have shown that they were concerned but uncertain on how to tackle antimicrobial resistance [36,37]. One of the studies also showed that while women like to be informed and make decisions for their health, they trust and place high importance on the opinion of the healthcare professional involved in their care [36]. The authors have used the insights gained from this audit to design a research study to explore prescribers’ decision-making practice for prescribing antibiotics to pregnant women. The barriers and facilitators to prescribing antibiotics according to guidelines will be identified to optimise the use of antibiotics in this population. The prescribing deviation from the guidelines seen in the results of this study is also valuable feedback for the prescribers. A pre and post analysis of prescribing patterns, once feedback is disseminated to the prescribers, is therefore also recommended for future work. In addition, future studies could also use a full year of data to mitigate monthly variation in prescribing patterns. This study also highlights the known gap in evidence for the optimal duration of antibiotic courses in pregnancy. Randomised controlled trials investigating shorter courses need to be conducted to provide evidence for effectiveness and safety as a step towards reducing antibiotics and promoting antimicrobial stewardship in antenatal care. At the same time, it is also important that patient safety is not compromised and therefore it is recommended that future work should investigate factors that prevent or delay antibiotic prescribing for UTIs in pregnant women. 

### 4.2. Limitations of the Study

The data were collected retrospectively by searching through patient records and using consecutive sampling to identify women who had been prescribed antibiotics in pregnancy. Although the search was conducted in consultation with an experienced user of the GP prescribing software, it is possible that some records were missed due to the way patients’ details are recorded and read codes assigned in the system. Inconsistency between the provision and documentation of self-care advice has been noted in previous research [38], and was also seen in this study, as it relied on prescribers documenting the provision of self-care advice in the medical records. Therefore, the results may not completely reflect the care that is provided during the consultation. The study was conducted to assess antimicrobial prescribing for UTIs in pregnancy. It used data from three London-based GP practices and had a small sample size, and therefore it may not be representative of the patient and healthcare professional demographic and prescribing practice in other regions of the world. A larger sample size is recommended in the future to improve the design and lead to a better powered study. This study, however, highlights important issues for antimicrobial prescribing in pregnant women which would be relevant to healthcare professionals involved in antenatal care and interested in safeguarding appropriate antibiotic use. 

## 5. Conclusions

The purpose of the study was to assess the appropriateness of primary care antimicrobial prescribing for UTIs in pregnancy. The results of the study highlight two important areas of concern. Firstly, the results show that the choice of first-line antibiotic is not optimal and second-line or non-recommended antibiotics are frequently prescribed to women. Secondly, self-care advice may need to be emphasised to women who are experiencing UTI symptoms during pregnancy. In conclusion, further work is required to review antibiotic use for UTIs in pregnancy and the provision of self-care advice, to facilitate a consolidated effort to optimise antibiotic use in response to the global threat of AMR. 

## Figures and Tables

**Table 1 medsci-08-00040-t001:** Data collection tool.

Demographic Data			Clinical Data			Treatment Data				
Age	Ethnicity	Trimester	Allergy	Symptoms	MSSU (Y/N)	Antibiotic	Dose	Duration	Self-care advice	Review (Y/N)

**Table 2 medsci-08-00040-t002:** Distribution according to ethnicity.

Ethnicity	Number of Records (*n* = 50)	Percentage (%)
White	21	42.0
Black	9	18.0
South Asian	7	14.0
Mixed	7	14.0
Other	6	12.0

**Table 3 medsci-08-00040-t003:** Antibiotic prescribing according to trimester of pregnancy.

Antibiotic	Number of Prescriptions in each Pregnancy Trimester		
	First	Second	Third
Nitrofurantoin	10	9	-
Amoxicillin	4	9	3
Cefalexin	2	6	3
Trimethoprim	1	-	-
Co-amoxiclav	1	2	-

**Table 4 medsci-08-00040-t004:** Proportion of antibiotic doses according to guideline.

Antibiotic	Dose		
	Optimum	High	Low
Nitrofurantoin	19	-	-
Amoxicillin	15	-	1
Cefalexin	4	5	2

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
