# Peer review of "Antibiotic Prescribing in Primary Care for Urinary Tract Infections (UTIs) in Pregnancy: An Audit Study"

_medsci, 2020, doi:10.3390/medsci8030040_

Round 1

Reviewer 1 Report

This study aimed to review the use of antibiotic therapy in UTIs in pregnant women in an outpatient setting. The goal of the study was to assess the prescribing patterns adherence to the National Institute for Health and Care Excellence guidelines.

1) The study would benefit from inclusion of higher number of patients. While a study of 50 patients has been previously published, which lead to the use of this number, increasing size would lead to a better powered study. Additionally, at least having a full year of data would also help mitigate differences in prescribing pattern by month, in case for some reason prescribing changed during Aug and Sept 2019.

2) It is not clear from the methodology if other antimicrobials such as 2nd and 3rd generation oral cephaloporins were included in the data collection. These agents are commonly used, at least in the US, for urinary tract infections and would be considered safe in pregnancy. It is also not clear if fluoroquinolones were included in the data collection. While these would not be considered safe in pregnancy, nor first-line due to resistance, prescribers occasionally do still prescribe them for urinary tract infections. 

3) Nitrofurantoin comes in two different salts, macrocrystal and macrocrystal/monohydrate, which differ in frequency of dosing q6h vs q12h, respectively. It is not clear which the patients were prescribed. Additionally, this can affect the assessment of frequency of dosing analysis that was done if prescribers used the wrong dosing frequency for a particular salt, which occurs in practice.

4) ~17% were prescribed antibiotics with doses that did not match guidelines, it would be nice to see the reasons added to a table, e.g., were the doses too high or low, etc.

5) This data demonstrating variation from guidelines would be valuable to the providers at these clinics. The authors may consider doing a pre/post analysis of prescribing patterns. If authors are planning on this, then it can be incorporated into the conclusion section as future direction.

6) The comment "Asymptomatic infections in non-pregnant women are not medically treated" - Line37, should state unless they are undergoing genitourinary surgery, in which case they would be treated.

Author Response

We would like to thank the reviewer for their comments and have carried out the relevant corrections.  A detailed response has been prepared, describing the amendments which are listed below. 

Review Report 1

This study aimed to review the use of antibiotic therapy in UTIs in pregnant women in an outpatient setting. The goal of the study was to assess the prescribing patterns adherence to the National Institute for Health and Care Excellence guidelines.

1) The study would benefit from inclusion of higher number of patients. While a study of 50 patients has been previously published, which lead to the use of this number, increasing size would lead to a better powered study. Additionally, at least having a full year of data would also help mitigate differences in prescribing pattern by month, in case for some reason prescribing changed during Aug and Sept 2019.

Thank you for your suggestion. We agree that increasing the sample size would lead to a better powered study and have therefore stated this on line 283. However, as mentioned, a study of 50 patients has been previously published and therefore this was the rationale employed in the current study. The aim of the study was to assess the prescribing practice and we did not focus on variation of the prescribing pattern by month, but we have recommended this for future work on line 264 as it would be an interesting observation. We collected data in July and limited the search from Oct 2018 to correspond with the time that the  NICE guideline was published, and we have now specified this on lines 108-110.

2) It is not clear from the methodology if other antimicrobials such as 2nd and 3rd generation oral cephaloporins were included in the data collection. These agents are commonly used, at least in the US, for urinary tract infections and would be considered safe in pregnancy. It is also not clear if fluoroquinolones were included in the data collection. While these would not be considered safe in pregnancy, nor first-line due to resistance, prescribers occasionally do still prescribe them for urinary tract infections. 

The antibiotics that were included were decided upon in consultation with the Chief Pharmacist at the health group and after reviewing NICE antimicrobial guidelines for UTI treatment. Only the antibiotics that are named in the method section were included in the search. Other 2nd or 3rd cephalosporins and fluoroquinolones were not included in the data collection as based on our experience and review of guidelines, these are not commonly used for UTIs in the UK, where this study was conducted. We have specified this on line 105 and line 283. 

3) Nitrofurantoin comes in two different salts, macrocrystal and macrocrystal/monohydrate, which differ in frequency of dosing q6h vs q12h, respectively. It is not clear which the patients were prescribed. Additionally, this can affect the assessment of frequency of dosing analysis that was done if prescribers used the wrong dosing frequency for a particular salt, which occurs in practice.

The patients should ideally, according to NICE guidelines, be prescribed the modified-release (MR) capsule which comes as 100mg with a q12h dosing. The rationale for this is cost-effectiveness and patient adherence rather than having any significant pharmacological implications. Therefore, as long as the dosing frequency for nitrofurantoin was correct according to the formulation prescribed, it was considered to be correct. Hence, both nitrofurantoin MR 100mg at q12h and nitrofurantoin 50mg q6h were deemed correct. Thank you for highlighting this and we have now specified this on Line 91 and Line 135.

4) ~17% were prescribed antibiotics with doses that did not match guidelines, it would be nice to see the reasons added to a table, e.g., were the doses too high or low, etc.

We have now included Table 4 (Line 162) to indicate how the doses did not match guideline with the reasons specified in the table.

5) This data demonstrating variation from guidelines would be valuable to the providers at these clinics. The authors may consider doing a pre/post analysis of prescribing patterns. If authors are planning on this, then it can be incorporated into the conclusion section as future direction.

Thank you for this recommendation. We have now included this on Line 261-264 as a direction for future work.

6) The comment "Asymptomatic infections in non-pregnant women are not medically treated" - Line 37, should state unless they are undergoing genitourinary surgery, in which case they would be treated.

Thank you, the statement has been amended as suggested on Line 38 and now reads ‘Asymptomatic infections in non-pregnant adults are not medically treated, unless they are undergoing genitourinary surgery…’.  

We would like to thank the reviewer again for their comments. With the changes suggested, we feel that the manuscript is now an improved piece of work.

Reviewer 2 Report

I think this is a very interesting and timely piece of work which contributes to discussions around ways of optimising treatment of UTIs in pregnant women in a way that supports the AMR agenda. I think this is a neat study and I enjoyed reading it. I have provided a few minor comments on the content:

Introduction

1.       Reference needed on line 27.

2.       Reference needed on line 28.

3.       The way lines 32-36 are written makes it look like asymptomatic bacteriuria requires antibiotic therapy for all. I know it’s across two paragraphs but is there someway that you can clarify that this is in pregnant women only or a caveat that asymptomatic bacteriuria in others like older adults do not require antibiotics? This is quite a contentious issue so I think it needs to be absolutely explicit.

a.       You could say in line 38 ‘ Asymptomatic infections in non-pregnant women should not be medically treated…’

Method

1.       In line 105, you might want to add something like ‘…pivmecillinam, which are not recommended but can be used to treat UTI in non-pregnant cases’.

2.       In lines 148-149 you say that there were no cases where antibiotics were not prescribed to pregnant women, but I thought your search only elicited pregnant women who had received an antibiotic therefore these women would not have been found in your search. Please clarify this.

Discussion

1.       You could mention that a self-care leaflet does exist for uncomplicated UTIs but can be used with pregnant women to promote self-care and preventative behaviours: LECKY DM, HOWDLE J, BUTLER C & MCNULTY CAM 2020. Women’s’ and general practitioners' experiences and expectations of the consultation for symptoms of uncomplicated urinary tract infection – a qualitative study informing the development of an evidence based, shared decision-making resource. British Journal of General Practice.

Author Response

We would like to thank the reviewer for their comments and have carried out the relevant corrections.  A detailed response has been prepared, describing the amendments which are listed below. 

Review Report 2

I think this is a very interesting and timely piece of work which contributes to discussions around ways of optimising treatment of UTIs in pregnant women in a way that supports the AMR agenda. I think this is a neat study and I enjoyed reading it. I have provided a few minor comments on the content:

Introduction

  1. Reference needed on line 27.

A reference has now been added on Line 27.

  1. Reference needed on line 28.

A reference has now been added on Line 28.

  1. The way lines 32-36 are written makes it look like asymptomatic bacteriuria requires antibiotic therapy for all. I know it’s across two paragraphs but is there some way that you can clarify that this is in pregnant women only or a caveat that asymptomatic bacteriuria in others like older adults do not require antibiotics? This is quite a contentious issue so I think it needs to be absolutely explicit.  You could say in line 38 ‘ Asymptomatic infections in non-pregnant women should not be medically treated…’

Thank you for pointing this out. We have re-arranged the sequence of sentences in this paragraph for clarity and included your suggestion on Line 37 which now reads ‘Asymptomatic infections in non-pregnant adults should not be medically treated, unless they are undergoing genitourinary surgery, whereas treatment of symptomatic UTIs consists of taking a short course of an antibiotic…’.

Method

  1. In line 105, you might want to add something like ‘…pivmecillinam, which are not recommended but can be used to treat UTI in non-pregnant cases’.

We have changed as suggested and now added that these are not recommended but can be used to treat UTIs in non-pregnant cases on Line 109.

  1. In lines 148-149 you say that there were no cases where antibiotics were not prescribed to pregnant women, but I thought your search only elicited pregnant women who had received an antibiotic therefore these women would not have been found in your search. Please clarify this.

Yes, thank you for pointing this out. This statement is indeed confusing, and we have deleted Line 154. Our search only elicited pregnant women who had received antibiotics and we would not have found any cases where antibiotics were not prescribed.

Discussion

  1. You could mention that a self-care leaflet does exist for uncomplicated UTIs but can be used with pregnant women to promote self-care and preventative behaviours: LECKY DM, HOWDLE J, BUTLER C & MCNULTY CAM 2020. Women’s’ and general practitioners' experiences and expectations of the consultation for symptoms of uncomplicated urinary tract infection – a qualitative study informing the development of an evidence based, shared decision-making resource. British Journal of General Practice.

Thank you for referring us to this article. It is a very useful resource and we have now mentioned it as a tool to use in consultations on Line 248.

We would like to thank the reviewer again for their comments. With the changes suggested, we feel that the manuscript is now an improved piece of work.

This manuscript is a resubmission of an earlier submission. The following is a list of the peer review reports and author responses from that submission.

Round 1

Reviewer 1 Report

Antibiotic prescribing in primary care for urinary tract infections (UTIs) in pregnancy: an audit study Thank you for having me reviewed this manuscript. Auditing antibiotic prescribing is a highly relevant issue and I’m not aware of an audit specifically focusing on UTI during pregnancy. The manuscript contains an interesting introduction. However, for the limited results, the Discussion section is rather long. I would propose to shorten the manuscript. Allow me to address the following issues with this manuscript: Major: - The authors have searched for pregnant women having received an AB prescription normally given for UTIs. They don’t seem to have incorporated UTIs without antibiotic prescribing which also would have been of importance in this audit study. Non-prescribing is not in accordance to guidelines either. Furthermore, when a non-expected non-recommend AB was prescribed, this was missed in the search too. - An audit of 50 records from 3 practices is rather limited. Minor: - Would have been interesting to know which 2nd choice antibiotics were prescribed. - Line 123: 36% did not comply to the duration of 7 days? Were they shorter or longer? In the Discussion, the answer is given, shorter, move to results. - Provision of self-care is probably under-estimated, as registration of this could have been omitted. - It could be interesting to add the reference: Willems et al. Fam Pract 2014; 31: 149, showing that women with cystitis need and value advice from their GP with respect to pain management. - In lines 163 it is explained why nitrofurantoin is contraindicated in the third trimester. Why haven’t the authors shown in the results which ABs were prescribed in which trimester? - Line 171: Have the authors checked for contraindication in the medical notes? It seems so, given the next sentence. But why not prescribed earlier in the results?

Author Response

Antibiotic prescribing in primary care for urinary tract infections (UTIs) in pregnancy: an audit study Thank you for having me reviewed this manuscript. Auditing antibiotic prescribing is a highly relevant issue and I’m not aware of an audit specifically focusing on UTI during pregnancy. The manuscript contains an interesting introduction. However, for the limited results, the Discussion section is rather long. I would propose to shorten the manuscript.

    Thank you for your comments. We have tried to make the discussion more concise but have also had to address comments from other reviewers. We have increased the manuscript length in line with the Assistant Editor’s comments by focusing on providing more detail in the Introduction, Method and Results section.

The authors have searched for pregnant women having received an AB prescription normally given for UTIs. They don’t seem to have incorporated UTIs without antibiotic prescribing which also would have been of importance in this audit study. Non-prescribing is not in accordance to guidelines either.

    We duly note that non-prescribing is not in accordance with the guidelines. We have not been able to assess this in our study as we did not come across such a case in our sample and we have now included a statement indicating this in the Results in line 159.  We also believe that this is an important area to explore for patient safety and have therefore suggested this in the future work section in the Discussion on line 262-265.

Furthermore, when a non-expected non-recommend AB was prescribed, this was missed in the search too.

    We did search for non-expected non-recommended antibiotics and have clarified this in the Methods section line 110. Our search included the following antibiotics; nitrofurantoin, amoxicillin, cefalexin, trimethoprim, co-amoxiclav, fosfomycin and pivmecillinam. Of these, trimethoprim, co-amoxiclav, fosfomycin and pivmecillinam are ones which are not recommended in pregnancy.

An audit of 50 records from 3 practices is rather limited.

We appreciate that this is a small sample size however it is in line with a similar audit that has been published previously and we have referenced this in the Methods section. We have also included this as a limitation of the study in the Discussion on line 280.

Would have been interesting to know which 2nd choice antibiotics were prescribed.

    We have clarified this in the Method section line 149. Amoxicillin and cefalexin are the 2nd choice antibiotics which were often prescribed instead of nitrofurantoin.

Line 123: 36% did not comply to the duration of 7 days? Were they shorter or longer? In the Discussion, the answer is given, shorter, move to results.

    Thank you for highlighting this. We have specified the shorter duration in the Results section too in line 169.

Provision of self-care is probably under-estimated, as registration of this could have been omitted.

    We agree with the reviewer on this point and our reliance on medical documentation for the study was included as a limitation in the Discussion. We have expanded this to clarify it is in relation to self-care advice and we have also discussed the inconsistency between provision and documentation with a reference to previous research on this in lines 268-269.

It could be interesting to add the reference: Willems et al. Fam Pract 2014; 31: 149, showing that women with cystitis need and value advice from their GP with respect to pain management.

    Thank you for bringing this reference to our attention. We have cited this reference within the Discussion in line 244 to support the view that women need and value advice from GPs.

In lines 163 it is explained why nitrofurantoin is contraindicated in the third trimester. Why haven’t the authors shown in the results which ABs were prescribed in which trimester?

    Thank you for highlighting this. We have now included this information in Table 3, which can be found in the Results section on page 4.

Line 171: Have the authors checked for contraindication in the medical notes? It seems so, given the next sentence. But why not prescribed earlier in the results? 

    We can confirm we checked for contraindications in the medical notes and have now amended the Results in lines 151-153 to make this clearer and also included this in the Method section in lines 114-117.

We would like to thank you for your comments and with the changes suggested, we feel that the manuscript is now an improved piece of work. 

Thank you for your consideration.

Reviewer 2 Report

Antibiotic Prescribing in Primary Care for Urinary Tract Infections (UTIs) in Pregnancy: An Audit Study.

The authors describe antibiotic prescribing patterns in pregnancy in 16 general practitioners at 3 large London-Based practices.

While disparities in actual practice patterns relative to current guidelines and the need for more care for pregnant women is a heartwarming message, this study adds nothing to the current literature. A 50-patient sample is hardly reflective of a practice pattern when it constitutes 0.14% (50/35,000) patients who had established care in these clinics. Make no mistake - cohorts of this size are potentially valuable in some areas of research, but absolutely not the way in which the authors are intending – particularly given the sparsity in clinical and provider characteristics reported. This manuscript has no external validity, such that it would be misleading so much as to imply that any of the authors’ observations apply to a reader’s own practice.

Author Response

While disparities in actual practice patterns relative to current guidelines and the need for more care for pregnant women is a heartwarming message, this study adds nothing to the current literature. A 50-patient sample is hardly reflective of a practice pattern when it constitutes 0.14% (50/35,000) patients who had established care in these clinics. Make no mistake - cohorts of this size are potentially valuable in some areas of research, but absolutely not the way in which the authors are intending – particularly given the sparsity in clinical and provider characteristics reported. This manuscript has no external validity, such that it would be misleading so much as to imply that any of the authors’ observations apply to a reader’s own practice.

Thank you for reviewing our manuscript and providing the review comments. We are mindful of the concerns expressed and appreciate that an audit study does not provide the level of external validity that is available with a larger research study. We would like to emphasise the benefit of an audit in highlighting discrepancies to be improved in clinical practice which this study has done in a specific setting based in London.  The results should therefore be interpreted within this context and to this end, we have cautioned the reader by specifying the issue of generalisability raised in the limitations section of the manuscript. We have highlighted the small sample size and the difference in practice in other UK regions in the Discussion in lines 278-281. We are keen to proceed with disseminating this study’s findings to the research community because we hope that it would encourage future national and global research interest in this area to benefit antimicrobial stewardship in antenatal care. The study provides potential direction for future work such as exploring prescribers’ decision-making for antibiotic prescribing and investigating factors that prevent or delay antibiotic prescribing.

Thank you for your consideration.

Reviewer 3 Report

The subject is extremely interesting and painful for medical community.

I recommend a few modification.

Please describe statistical methods that you used, it's not clear described.

The conclusion need to clear and specific. I recommend three short conclusion.

Please recheck the References order.

Thanks for the opportunity of reading the article.

Author Response

The subject is extremely interesting and painful for medical community.

I recommend a few modification.

Please describe statistical methods that you used, it's not clear described.

    Thank you for highlighting this. We have clarified our use of descriptive statistics to collate and provide a summary in the Results on line 119-120.

The conclusion need to clear and specific. I recommend three short conclusion.

    Thank you for the comment. We have tried to be more clear and concise in the Conclusion by using three points. We initially signpost the reader to two main results and in our third point we explain the significance of these.

Please recheck the References order.

    We have re-checked the references order after citing additional references.

We would like to thank you for your comments and with the changes suggested, we feel that the manuscript is now an improved piece of work. 

Thank you for your consideration.

Round 2

Reviewer 1 Report

Thank you for incorporating and changing text in the manuscript based on the recommendations. The results are clearly described now and all described in the results section. The number of consultation included in this audit is limited, as highlighted earlier, and up to the editor whether this fulfills the criteria of the journal. The paper is interesting and pleasant reading, describing an under-researched topic of antibiotic prescribing quality in primary care.

Author Response

Thank you for reviewing our reply and for confirming you are happy with the changes made. We can confirm that previous published work has had a similar sample size (Mosedale, T.; Kither, H.; Byrd, L. PM.12 The Management of Pregnant Women Attending Triage with Suspected Urinary Tract Infection. Arch Dis Child Fetal Neonatal Ed 2013, 98, 1, A29) and that Pharmacy journal has previous published audits (Dobia, A.; Ryan, K.; Grant, D.; BaHammam, A. Current Clinical Practice for the Use of Hypnotics to Manage Primary Insomnia in Adults in a Tertiary Hospital in Saudi Arabia: An Audit Study. Pharmacy 2019, 7, 15).

Reviewer 2 Report

The authors said the audit is specific to their own practice. That is a quality improvement project, not a publishable study.

“The study provides potential direction for future work such as exploring prescribers’ decision-making for antibiotic prescribing and investigating factors that prevent or delay antibiotic prescribing.“ - I unfortunately believe Their study design supports these claims, and would go so far as to say it can have consequences when readers try and interpret this as it relates to their own practice .

Author Response

Thank you for reviewing our reply. We can confirm that the Pharmacy journal has published audit studies previously (Dobia, A.; Ryan, K.; Grant, D.; BaHammam, A. Current Clinical Practice for the Use of Hypnotics to Manage Primary Insomnia in Adults in a Tertiary Hospital in Saudi Arabia: An Audit Study. Pharmacy 2019, 7, 15.) and we hope our work can also be published to add to the literature on antibiotic prescribing for urinary tract infections in pregnancy.